# Group and Individual Changes in Spinal Mobility During a 12-Week Rehabilitation Program Including Swimming in Horses with Axial Musculoskeletal Lesions

**DOI:** 10.3390/ani16010103

**Published:** 2025-12-30

**Authors:** Baptiste Pécresse, Claire Moiroud, Sandrine Hanne-Poujade, Chloé Hatrisse, Emeline De Azevedo, Virginie Coudry, Sandrine Jacquet, Fabrice Audigié, Henry Chateau

**Affiliations:** 1ACAP3, Ecole Nationale Vétérinaire d’Alfort, F-14430 Goustranville, France; baptiste.pecresse@vet-alfort.fr (B.P.);; 2Labcom LIM-ENVA, LIM Group, F-24300 Nontron, France; 3Ecole Nationale Vétérinaire d’Alfort, F-94700 Maisons-Alfort, France

**Keywords:** aquatic training, axial disorders, back pain, equine rehabilitation, inertial measurement units, swimming, spinal mobility, thoracolumbar kinematics

## Abstract

Back pain and spinal disorders often impair performance and lead to early retirement in sport horses. Swimming is commonly included in rehabilitation programs because water supports body weight, reducing limb stress while preserving cardiovascular fitness and muscle activity. However, its effect on spinal mobility remains poorly quantified. This study involved sixteen sport horses diagnosed with cervical or thoracolumbar musculoskeletal lesions who completed a 12-week rehabilitation program, including swimming sessions. Inertial sensors were placed along the spine to record weekly measurements while horses trotted in a straight line on a hard surface. Group-level analyses revealed only limited changes in back mobility across the different phases of the program. Although some horses showed individual variations, these were not consistent enough to alter the overall interpretation. Together, these findings indicate that swimming does not produce major, uniform changes in thoracolumbar mobility and underline the importance of monitoring each horse individually throughout rehabilitation, whether or not aquatic exercise is included, to guide clinical decisions and optimize recovery.

## 1. Introduction

Locomotor disorders, along with suboptimal performance, are the leading causes of premature retirement in race and sport horses [1,2,3]. Their management requires an accurate diagnosis [4] and the implementation of tailored therapeutic strategies aimed at promoting rapid recovery and return to prior performance levels [5].

Despite requiring specific infrastructure, equine swimming pools have seen widespread adoption since the 1970s [6,7,8,9,10]. The physical properties of water offer several therapeutic benefits. Buoyancy reduces axial loading on the limbs, and the movements performed during swimming can increase joint excursions, with most joints exhibiting a greater range of motion during swimming than in trot [11]. Aquatic exercise is therefore considered a way to maintain training while minimizing joint and soft tissue stress, thereby facilitating an earlier return to activity [12]. Water resistance promotes muscle recruitment and enhances neuromuscular control [12,13], while also stimulating the cardiorespiratory system [14] and improving venous and lymphatic return [15]. As such, swimming is now widely incorporated into musculoskeletal rehabilitation [14,16] and conditioning protocols [17,18].

However, swimming requires horses to keep their heads above the waterline, which is believed to increase hyperextension of the cervical, thoracolumbar, and pelvic spinal regions [12]. This biomechanical posture may contraindicate aquatic work in horses with axial pathologies, particularly of the cervical spine [12,16].

Dorsal mobility can be assessed using clinical scoring systems [19,20]. Recent work has compared water treadmill, dry treadmill, and lunging exercise protocols over a 6-week rehabilitation period in horses with primary back pain. The study reported increased mechanical nociceptive thresholds and enhanced lumbar muscle scores but highlighted the absence of objective kinematic analyses to complement these clinical assessments [21]. Spinal mobility can be quantified through instrumented approaches such as optical motion capture (MOCAP) with skin markers [22,23] or inertial measurement units (IMUs) [24,25]. A recent validation study demonstrated agreement between MOCAP- and IMU-based measurements of flexion–extension at the thoracolumbar junction, with no proportional bias [24]. Given their ease of use in the field, IMUs are particularly well suited for longitudinal monitoring. The present study therefore applied the IMU-based methodology developed and validated by Hatrisse et al. (2023) [24].

To date, no study has specifically evaluated the effect of swimming exercise on dorsal mobility in horses with axial disorders. Therefore, the present work aimed to characterize the changes over time in thoracolumbar flexion–extension, measured during straight-line trot, in horses diagnosed with at least one axial disorder and enrolled in a standardized training program including swimming sessions. The primary objective was to assess changes in thoracolumbar flexion–extension over time and investigate whether these changes differ depending on lesion location.

This study aimed to characterize changes over time in thoracolumbar flexion–extension range of motion during a 12-week rehabilitation program including swimming in horses with axial musculoskeletal lesions. Analyses were stratified by lesion location (cervical vs. thoracolumbar) in order to describe longitudinal changes within each group separately, without formally testing for differences between groups.

We hypothesized that thoracolumbar flexion–extension range of motion would remain broadly stable across training phases, with no statistically significant group-level differences between the land and aquatic training periods within each lesion group.

## 2. Materials and Methods

### 2.1. Animals Included in the Study

This prospective study was conducted at CIRALE—Locomotor Pathology Unit of the Equine Veterinary Teaching Hospital at the National Veterinary School of Alfort (EnvA), starting in December 2022. Seventeen horses aged between 5 and 14 years (mean age ± standard deviation SD: 9.5 ± 2.6 years, mean height ± SD: 167 ± 7 cm, mean body mass ± SD: 555 ± 51 kg) were initially enrolled. All horses had been regularly trained for show jumping or eventing for at least a year before admission. While some were actively competing, others followed structured jumping work as part of their routine, despite not participating in official competitions. The sample consisted of 12 geldings and 5 mares. All horses had a documented history of work-related difficulties or poor performance attributed to back or neck pain.

The primary inclusion criterion was the presence of radiographic abnormalities affecting the cervical and/or thoracolumbar spine. Horses with isolated sacroiliac lesions without concurrent cervical or thoracolumbar involvement were not included. Horses presenting with limb-related locomotor disorders causing overt lameness were excluded. Mild, clinically stable locomotor asymmetries compatible with regular exercise, as commonly encountered in a clinical rehabilitation context, were allowed. Horses were also excluded if they failed to tolerate pool work after a 3- to 4-session acclimatization period, or if clinical deterioration occurred during the protocol requiring substantial modification of the training program and/or anti-inflammatory treatment for more than five consecutive days. One horse was excluded due to myositis and skin lesions, resulting in a final cohort of 16 horses (Table 1). Prior to the procedure, the protocol was examined and approved by the dedicated clinical research ethics committee (Comité d’Ethique en Recherche Clinique, EnvA, No. 2022-09-19).

### 2.2. Grouping Based on Lesion Location

The 16 horses were assigned to two groups according to the predominant region of axial involvement on diagnostic imaging, following the classification of Audigié et al. [26] (Table 1).

The cervical group included four horses exhibiting radiographic abnormalities primarily affecting the caudal cervical vertebrae (beyond C5), all with signs of osteoarthritis changes of the articular processes-synovial intervertebral joints (AP-SIVJ). Three of these horses also showed evidence of intervertebral disc degeneration at the same sites (vertebral symphysis abnormalities).

The thoracolumbar group consisted of twelve horses with predominant thoracic and/or thoracolumbar and/or lumbo-sacral abnormalities of different grades of severity (mild to severe). Imaging findings were as follows: osteoarthritic changes of the articular processes-synovial intervertebral joints (AP-SIVJ) (9 horses), impingement of the dorsal spinous processes (7 horses), abnormalities of the curvature of the vertebral axis (lordosis, kiphosis, scoliosis) (3 horses), middle thoracic ventral spondylosis (2 horses), lesion of the lumbo-sacral symphysis (lumbo-sacral intervertebral disc degeneration in one horse), and vertebral body dysplasia (1 horse).

### 2.3. Training Protocol

All included horses followed a standardized two-phase training protocol designed to ensure controlled progression and enable longitudinal monitoring of changes in thoracolumbar mobility under consistent training conditions. Throughout the entire protocol, all horses were placed in a standard motorized horse walker for 30 to 45 min each morning before working and allowed to spend a few hours per day moving freely in a small paddock in the afternoon.

Phase 1, referred to as the land training (LT) phase, lasted four weeks and consisted exclusively of land-based ridden work (Table 2). It aimed to help horses acclimate to their new environment and riders. Two riders were responsible for the entire group, and all horses followed the same weekly schedule in terms of exercise type (flatwork, gymnastic-cavaletti, jumping, gallop on a track, and lunge work). The intensity of each session (duration, speed, fence height) was adapted to the individual horse’s level, based on its usual workload prior to admission. For horses with reduced training before inclusion, intensity was progressively increased over the first two weeks. From week 3 onwards, training intensity was maintained at a consistent level for all horses until the end of the protocol. All ridden sessions began with a standardized warm-up consisting of 15 min of walk followed by 4 min of gallop (2 min at each lead), prior to 25–30 min of work. This phase also served to establish baseline measurements of thoracolumbar mobility for each horse.

Phase 2, referred to as the aquatic training (AT) phase, spanned the following eight weeks. During this phase, swimming sessions were progressively incorporated into the training routine. Three ridden sessions per week were replaced by three swimming sessions (Table 2).

Swimming sessions were performed in a purpose-built equine swimming pool similar to that previously described by Gaulmin et al. [10]. The pool featured a U-shaped layout, approximately 55 m in length, allowing horses to enter and leave the water on each lap. Water depth was 3 m, ensuring full flotation and preventing any ground contact throughout a swimming lap. Horses were guided using two lunge lines held by experienced handlers positioned on either side of the animal, and all sessions were performed under continuous supervision. To preserve natural swimming behavior, horses were allowed to swim at a self-selected pace.

Phase 2 was divided into two consecutive four-week sub-periods: Aquatic training 1 (AT1) and Aquatic training 2 (AT2). This subdivision was designed to allow a more detailed temporal analysis of the variables of interest, as swimming workloads were progressively increased throughout AT1 by increasing the number of laps performed per session (Table 3). During week 5 (the first week of swimming), horses swam on a lap-by-lap basis, with an inter-lap recovery period of approximately 45–60 s to allow respiratory rate normalization before re-entering the pool. From week 6 onwards, each swimming session began with a standardized warm-up consisting of one lap in each direction, separated by a recovery period of approximatively 45 s. Thereafter, horses performed interval swimming, organized into fractions comprising successive laps swum continuously (horses exited and re-entered the pool immediately between laps, without rest), followed by a 3 min period of active recovery at walk on a treadmill inclined at 3°. Each week included one session with two fractions and two sessions with three fractions, with a progressive increase in the number of laps per fraction. The most demanding sessions consisted of two warm-up laps followed by three fractions of five consecutive laps. The direction of turns was alternated at each fraction to balance laterality effects.

### 2.4. Data Acquisition

The range of motion (ROM) at the thoracolumbar junction was used as the primary indicator of dorsal mobility in this study (T18 angle). The method followed the protocol described by Hatrisse et al. (2023) [24], involving straight-line trot recordings over a hard 25 m track. At the end of each week, horses performed at least one out-and-back sequence (i.e., a minimum of 50 m in total) at the trot. All assessments were conducted in-hand by an experienced handler, and identical equipment (bridle and lead rope) was used for all horses to ensure consistency across recordings. Speed was not controlled during these recordings; horses trotted at their spontaneous comfortable speed. To account for potential differences in locomotor rhythm across sessions, stride frequency was quantified for each stride and later included as a covariate in the statistical models. Measurements were performed once per week throughout the 12-week study period.

Three synchronized IMUs (Blue Trident^®^, Vicon Motion Systems Ltd., Yarnton, UK) were mounted on the median dorsal line at the withers, the 18th thoracic vertebra (T18), and at the level of the tubera sacrale. Each sensor recorded data at a sampling rate of 225 Hz, with a measurement range of ±16 g, a maximum angular velocity of 2000°/s, and 16-bit resolution.

### 2.5. Data Processing

IMU data were processed according to the framework validated by Hatrisse et al. [24]. Briefly, raw accelerations were first reoriented into a terrestrial reference frame using a static standing trial to correct sensor orientation relative to gravity. Vertical displacement curves were then obtained by double integration of the corrected accelerations, combined with appropriate filtering. For each horse, the horizontal distances between the withers–T18 and T18–pelvis sensors were measured manually on the standing horse and incorporated into the trigonometric model to account for individual variation in back length.

The thoracolumbar flexion–extension angle (T18 angle) was computed from these displacement signals using Matlab (R2021b, The MathWorks Inc., Natick, MA, USA), then filtered to remove high-frequency noise [27]. The thoracolumbar flexion–extension angle exhibits two flexion peaks and two extension peaks within each stride cycle (Figure 1). These extrema correspond, respectively, to moments of minimal and maximal dorsal excursion during stance and swing phases. For each stride *i*, range of motion of the two flexion movements (Flexion1, Flexion2) and two extension movements (Extension1, Extension2) were extracted.

To obtain a stride-level descriptor of sagittal-plane back motion, the two flexion ranges of motion and the two extension ranges of motion were averaged within each stride:(1)$${\mathrm{Flexion}}_{\mathrm{Stride}(i)}=\frac{{\mathrm{Flexion1}}_{\mathrm{Stride}(i)}+{\mathrm{Flexion2}}_{\mathrm{Stride}(i)}}{2}$$(2)$${\mathrm{Extension}}_{\mathrm{Stride}(i)}=\frac{{\mathrm{Extension1}}_{\mathrm{Stride}(i)}+{\mathrm{Extension2}}_{\mathrm{Stride}(i)}}{2}$$

Mean flexion and extension ranges of motion were then calculated across all strides of an examination:(3)$${\mathrm{Flexion}}_{\mathrm{Exam}}=\mathrm{mean}({\mathrm{Flexion}}_{\mathrm{Stride}(i)})$$(4)$${\mathrm{Extension}}_{\mathrm{Exam}}=\mathrm{mean}({\mathrm{Extension}}_{\mathrm{Stride}(i)})$$

The final thoracolumbar mobility descriptor, the mean T18 flexion–extension range of motion (ROM), was defined as:(5)$${\mathrm{ROM}}_{\mathrm{Exam}}=\frac{{\mathrm{Flexion}}_{\mathrm{Exam}}+{\mathrm{Extension}}_{\mathrm{Exam}}}{2}$$

Baseline locomotor symmetry indices (vertical displacement asymmetry at the withers and pelvis) were extracted from the same IMU recordings at inclusion for descriptive purposes. These indices were expressed as percentages according to the EQUISYM^®^ methodology (EQUISYM^®^, LIM Group, Nontron, France).

### 2.6. Statistical Analysis

To assess changes in mean T18 flexion–extension range of motion, a linear mixed-effects model (LME) was fitted separately for the cervical and thoracolumbar groups. The dependent variable was the mean T18 flexion–extension range of motion per stride, measured during straight-line trot. Fixed effects included training phase (LT, AT1, AT2), week (treated as a continuous variable ranging from 1 to 4 within each phase), and stride frequency (in strides per second). This last adjustment aimed to account for possible differences in locomotor rhythm between training phases, given that speed was not directly controlled during data collection.

Random effects were hierarchically structured to account for both inter- and intra-individual variability: Horse was included as a random intercept, with Stride nested within Horse to account for repeated measurements.

Model assumptions were verified through visual inspection of residual plots: normality was assessed using Q-Q plots, and homoscedasticity was verified through standardized residual analysis. To further explore within-subject patterns, a second set of LMEs was built for each individual horse, using the same fixed effects but including only Stride as a random effect. LMEs are well suited for longitudinal analyses and allow handling of missing-at-random data, where applicable [28].

Differences in mean T18 flexion–extension range of motion between training phases (main effect) were tested using analysis of variance (“type III” ANOVA). When the main effect was statistically significant (*p* < 0.05), pairwise comparisons were conducted using Tukey–Kramer post hoc tests [29]. Effect sizes (Cohen’s |d|) were calculated to evaluate the clinical relevance of observed differences, based on conventional thresholds [30]: d < 0.2 (negligeable), 0.2 ≤ d ≤ 0.5 (small), 0.5 ≤ d ≤ 0.8 (medium), d > 0.8 (large).

All statistical analyses were performed using R software version 4.4.2 (R Core Team, R Foundation for Statistical Computing, Vienna, Austria), including the stats [31] and lmerTest packages [32].

## 3. Results

### 3.1. Descriptive Analysis of the Dataset

A total of 5763 strides were recorded from the 16 horses included in the study. Six locomotion examinations were excluded from analysis due to acquisition failures: two for horse #02 (week 5 and week 12), two for horse #19 (week 3 and week 9), and one each for horses #13 (week 4) and #14 (week 11). All exclusions were made prior to statistical processing to ensure consistency and validity of the dataset.

On average, each locomotion examination yielded 15.8 ± 9.2 strides for horses in the cervical group and 16.7 ± 9.8 strides for those in the thoracolumbar group.

The mean stride frequency was 1.38 ± 0.22 strides/s (std/s) in the cervical group and 1.41 ± 0.19 std/s in the thoracolumbar group.

Baseline movement symmetry data for all horses are reported in Appendix A. These values provide a descriptive overview of the locomotor symmetry status of the study population at inclusion.

Across the dataset (Table 4), mean T18 flexion–extension range of motion ranged from approximately 2° to 5° depending on individual variability and lesion type. Horses with cervical lesions generally exhibited slightly higher values than those with thoracolumbar lesions, although this observation should be interpreted cautiously given the small number of cervical cases.

### 3.2. Group Changes in Thoracolumbar Mobility Across Training Phases

Results of the LME analyses are summarized in Table 5, while detailed type III ANOVA outputs for the effects of phase, week, and stride frequency are presented in Appendix B. In the cervical group (n = 4 horses, 1271 strides), no significant variation in T18 flexion–extension was observed across the three training phases (*p* = 0.092). Conversely, the thoracolumbar group (n = 12 horses, 4492 strides) showed a significant main effect of training phase (type III ANOVA: *p* < 0.001). Post hoc Tukey–Kramer tests revealed no significant difference in T18 flexion–extension range of motion between LT and AT1 phases but a small yet significant decrease between AT2 and LT (−0.1°, 95% CI: [−0.1; 0], *p* = 0.002) and between AT2 and AT1 (−0.1° [−0.1; 0], *p* < 0.001). However, the corresponding Cohen’s d values (0.1 and 0.2, respectively) indicated negligible and small effect sizes according to conventional thresholds.

### 3.3. Individual Changes in Thoracolumbar Mobility Across Training Phases in Horses with Cervical Lesions

To illustrate the heterogeneity of responses within the cervical lesion group, individual linear mixed-effects analyses were performed for each horse. These analyses were intended to describe within-horse temporal patterns and were not designed to support group-level inference.

As reported in Table 5, all four horses with cervical lesions showed a significant main effect of training phase (type III ANOVA, *p* < 0.05), indicating that thoracolumbar flexion–extension range of motion varied over time within individuals. However, the direction and magnitude of these changes were inconsistent across horses. The corresponding individual changes over time are shown in Appendix C, Figure A1a and Figure A2a.

Two horses (#02 and #19) exhibited significant decreases in range of motion during the AT2 phase compared with LT, with absolute changes of −0.3° and −0.7°, corresponding to relative decreases of 7% and 16%, respectively. Effect sizes were small for horse #02 (Cohen’s d = 0.3) and medium for horse #19 (d = 0.8). In contrast, two horses (#06 and #14) showed significant increases in range of motion during AT2 compared with LT (+0.1° and +0.5°, corresponding to relative increases of 3% and 10%), both associated with small effect sizes (d ≤ 0.4).

Overall, although statistically significant phase-related changes were detected at the individual level, their heterogeneous direction and limited magnitude preclude any consistent interpretation regarding the effect of aquatic training in horses with cervical lesions.

### 3.4. Individual Changes in Thoracolumbar Mobility Across Training Phases in Horses with Thoracolumbar Lesions

Individual analyses were also conducted for horses in the thoracolumbar lesion group to characterize interindividual variability in longitudinal responses. The corresponding individual changes over time are shown in Appendix C, Figure A1b and Figure A2b.

As shown in Table 5, five horses (#03, #11, #12, #13, and #23) did not exhibit any significant change in mean T18 flexion–extension range of motion over the 12-week protocol (type III ANOVA, *p* > 0.05).

Seven horses showed a significant main effect of training phase (*p* < 0.05), indicating that thoracolumbar mobility varied over time within these individuals. Among them, four horses (#01, #17, #18, and #20) demonstrated significant decreases in range of motion during AT2 compared with LT, with absolute reductions ranging from −0.2° to −0.8° (relative decreases of 8% to 18%). Effect sizes were small for one horse (d = 0.4) and large (d > 0.8) for three horses. In contrast, three horses (#09, #15, and #21) exhibited significant increases in range of motion during AT2 compared with LT, with absolute changes between +0.1° and +0.3° (relative increases of 3% to 14%) and small to medium effect sizes.

No horse showed a monotonic or uniform pattern of increase or decrease across all three phases. Moreover, the magnitude of individual changes rarely exceeded 1°, remaining close to the resolution limits of the measurement system.

Taken together, these individual analyses highlight marked inter-horse variability in thoracolumbar mobility trajectories within the thoracolumbar lesion group. These patterns are presented for descriptive purposes only and should not be interpreted as evidence of consistent or predictable biomechanical effects of aquatic training at the individual level.

## 4. Discussion

This study provides the first longitudinal, sensor-based quantification of thoracolumbar mobility in horses with axial musculoskeletal lesions undergoing a standardized rehabilitation program that included swimming. The main findings were (1) consistently greater thoracolumbar motion in horses with cervical lesions compared with those with thoracolumbar lesions and (2) limited group-level changes over the 12-week program despite marked interindividual variability in mobility trajectories.

### 4.1. Group-Level Findings and Biomechanical Interpretation

Horses with cervical lesions systematically exhibited greater T18 flexion–extension range of motion than horses with thoracolumbar lesions. Although this trend aligns with previous kinematic findings suggesting lesion-location-dependent motion patterns, the small number of cervical cases and the lack of adjustment for stride frequency preclude any definitive conclusion. This observation should therefore be considered exploratory and warrants confirmation in larger, balanced cohorts. Such differences, if confirmed, could reflect compensatory adaptations of trunk motion to altered head–neck posture, as suggested by Gómez Álvarez et al. [33] and Rhodin et al. [34].

The absolute range of motion values measured in this study align with those reported by Hatrisse et al. [24] in sound horses (approximately 4–5°). Horses with thoracolumbar lesions generally remained below these reference ranges, even after aquatic training. Although this observation does not allow causal inference, it suggests that reduced thoracolumbar motion could be a sensitive indicator of axial stiffness or discomfort. Interpreting reductions in thoracolumbar flexion–extension range of motion at the trot, however, requires caution. From a biomechanical standpoint, trot is characterized by dynamic stability arising from diagonal limb support, which naturally constrains sagittal-plane excursion. Within this context, a decrease in range of motion cannot be straightforwardly considered either beneficial or detrimental. Clinically, horses with axial musculoskeletal pain often exhibit reduced spinal mobility, consistent with a protective stiffening strategy intended to minimize loading on painful structures. Objective kinematic studies support this interpretation: Wennerstrand et al. [35] showed that sport horses with clinical back pain displayed smaller thoracolumbar flexion–extension ranges of motion during trot compared with asymptomatic controls, suggesting that diminished dorsal excursion may reflect discomfort or functional impairment. This reinforces the clinical relevance of incorporating objective back mobility metrics into locomotor evaluations, particularly for horses classified as “dorsalgic”.

### 4.2. Within-Group Changes over Time and Temporal Patterns

When examined within groups, only the thoracolumbar group exhibited a significant phase effect, characterized by a decrease in thoracolumbar mobility during the second four-week aquatic phase (AT2), whereas no significant difference was observed at the end of the first aquatic phase (AT1). This decrease remained very limited in magnitude, corresponding to an average reduction of approximately 3% in thoracolumbar mobility, with associated Cohen’s d values of 0.2, indicating negligible to small effect sizes according to conventional thresholds.

The subsequent decrease in range of motion observed during AT2 may reflect either enhanced trunk stability or reduced compensatory movement. However, group-level analyses do not account for the substantial interindividual variability observed, which warrants examination of individual mobility trajectories.

### 4.3. Individual Variability and Clinical Relevance

Although individual horses displayed heterogeneous adaptations in thoracolumbar flexion–extension range of motion over the 12-week program, these trajectories were not intended to support group-level conclusions. Instead, they illustrate the variability commonly encountered in horses affected by axial musculoskeletal disorders and undergoing rehabilitation in clinical settings.

At the individual level, both increases and decreases in thoracolumbar mobility were observed, with no consistent direction or temporal pattern across horses, regardless of lesion location. Importantly, the absolute magnitude of these changes was generally small and rarely exceeded 1°, remaining close to the effective angular resolution of the IMU-based measurement system. This finding reinforces the need for caution when interpreting statistically significant individual effects, particularly in the absence of a control group and given the inherent biological and measurement variability.

From a clinical perspective, this heterogeneity likely reflects the multifactorial nature of axial musculoskeletal dysfunction, where biomechanical responses may be influenced by lesion type, anatomical location, chronicity, baseline neuromuscular control, and individual adaptation strategies. As a result, aquatic training should not be viewed as a uniformly beneficial or detrimental intervention for thoracolumbar mobility. Rather, its effects appear to be highly individual and context-dependent.

These results underscore the importance of individualized, longitudinal monitoring in equine rehabilitation, whether or not swimming is included in the training program. Objective kinematic assessments using inertial sensors provide a practical tool to quantify changes in spinal motion over time, helping clinicians identify meaningful deviations from an individual baseline and adapt rehabilitation strategies accordingly. However, such measurements should be interpreted within a broader clinical and biomechanical framework and not in isolation or as indicators of expected group-level effects.

### 4.4. Methodological Considerations

Several methodological limitations must be acknowledged. First, the relatively small sample size, particularly in the cervical group, limits the statistical power and generalizability of these findings. The absence of a non-swimming control group also prevents definitive attribution of observed changes to aquatic exercise alone. Moreover, mobility was assessed only during straight-line trot on a hard surface, which may not fully capture spinal dynamics during canter, ridden work or on curved paths. Although this gait provides more standardized and reproducible locomotor conditions across horses and over time, thereby facilitating longitudinal comparisons, thoracolumbar flexion–extension range of motion at the trot remains relatively limited due to the dynamic stability associated with diagonal limb support. Consequently, incorporating the canter, which exhibits more pronounced sagittal-plane spinal excursions, will be necessary in future studies to capture the full dynamic range of thoracolumbar mobility and to refine the interpretation of axial musculoskeletal adaptations.

From a technical standpoint, the kinematic analysis was restricted to a single angular descriptor (T18 angle), providing a unidimensional view of dorsal motion and excluding lateral bending and axial rotation. While this parameter has proven repeatable and biomechanically meaningful [24], it represents only a global measure of thoracolumbar mobility and does not account for regional variations. Measurement accuracy is also limited by IMU resolution, potential sensor misalignment, and soft-tissue artifacts. The effective angular resolution of the system, estimated between 0.2° and 0.5°, implies that subtle variations may fall below the detection threshold, particularly when inter-session variability is high. Furthermore, IMUs record relative angular displacement but cannot distinguish between active neuromuscular control and passive mechanical motion.

Although horse speed was not directly measured during the recordings, stride frequency (cadence) was used as a proxy indicator of locomotor rhythm and included as a covariate in all statistical models. However, stride frequency does not entirely replace direct speed measurement, as it does not account for stride length variations. By adjusting for stride frequency, we ensured that the differences observed in thoracolumbar motion could not be attributed to changes in locomotor rhythm, thereby strengthening the interpretation of training-related effects.

Beyond its role as a statistical adjustment, stride frequency emerged as an important methodological factor in the interpretation of back kinematics at the trot. Even though cadence varied only minimally in our dataset, it showed a significant association with thoracolumbar flexion–extension range of motion. This finding highlights the necessity of systematically accounting for locomotor rhythm in studies of spinal biomechanics. The restricted range of cadence values in our study, however, limits the extent to which this relationship can be fully explored. Future work performed across a broader range of speeds will be essential to better characterize the interaction between stride frequency and thoracolumbar mobility.

In addition to these methodological considerations, it is important to relate our findings to existing knowledge on the neuromuscular control of the equine back. Electromyographic studies have shown that the longissimus dorsi and multifidus muscles play a key role in stabilizing the thoracolumbar region during locomotion, with activation patterns that vary across gaits and phases of the stride [36,37,38]. These muscles contribute to both dynamic stability and modulation of sagittal-plane motion, suggesting that changes in flexion–extension range of motion may reflect not only mechanical factors but also neuromuscular strategies.

Furthermore, several multi-segment kinematic studies have quantified spinal motion across multiple vertebral landmarks, demonstrating that thoracolumbar mobility is influenced by gait, speed, posture, and training conditions [23,39,40]. Together, these studies highlight the complexity of dorsal motion and support the need to interpret IMU-based measurements within a broader biomechanical framework. While the single-angle descriptor used in the present study provides a robust and field-applicable metric, integration with complementary approaches (such as EMG, ultrasonography of epaxial muscles, or multi-level motion capture) will be essential to deepen the understanding of axial musculoskeletal adaptations in future studies.

### 4.5. Clinical Implications and Future Directions

From a clinical standpoint, these results suggest that swimming, when incorporated into a structured rehabilitation program, does not induce substantial changes in thoracolumbar mobility at the group level. However, the marked interindividual variability observed (characterized by both increases and decreases in range of motion) highlights the need for individualized monitoring, whether or not the rehabilitation program includes swimming. Objective kinematic assessments performed periodically throughout training may help clinicians detect meaningful changes in spinal function and adjust exercise intensity or duration accordingly. By including stride frequency in the model, we ensured that the reported differences in back motion were independent of locomotor rhythm. This emphasizes the value of combining kinematic analysis with precise temporal gait descriptors when evaluating rehabilitation effects.

While Geiger et al. (2025) [21] reported clinical improvements in horses undergoing unridden rehabilitation based on scoring systems and pressure algometry, their evaluations did not include objective motion analyses. In contrast, the present study provides a quantitative, sensor-based characterization of thoracolumbar dynamics, allowing for a more detailed and individualized evaluation of functional changes during rehabilitation. This complementary approach enhances the understanding of locomotor adaptation beyond clinical scoring alone.

The moderate to large effect sizes observed in some individuals indicate that thoracolumbar motion parameters may serve as sensitive markers of functional adaptation, even when absolute range of motion changes remain small. Whether such kinematic variations reflect beneficial neuromuscular adjustments or compensatory mechanisms associated with lesion chronicity remains to be determined. Moreover, clinical expression may vary even in the absence of structural lesion progression, for instance due to muscular pain, contractures, or fluctuations in the inflammatory activity of the lesion.

Repeated imaging sessions (including radiography, ultrasonography, and scintigraphy) were scheduled as part of the broader study protocol, together with detailed clinical examinations assessing spinal mobility, vertebral and muscular sensitivity, and muscular condition, as well as dynamic scoring during locomotion. A dedicated analysis of these data is currently underway and will help clarify the relationships between lesion dynamics, pain expression, and kinematic changes. These complementary results will be reported in forthcoming publications, thereby extending and consolidating the findings of the present study.

Future research should aim to identify factors influencing responsiveness to aquatic training, such as baseline spinal mobility, lesion type, or neuromuscular asymmetries. Combining IMU-based kinematic analysis with complementary modalities (such as electromyography, muscle ultrasonography, or advanced musculoskeletal modelling) would provide a more comprehensive understanding of axial function and compensatory strategies. Ultimately, integrating these multimodal assessments could refine rehabilitation strategies and support more personalized management of horses with axial musculoskeletal disorders.

## 5. Conclusions

This study investigated changes in thoracolumbar mobility at the trot in horses with axial musculoskeletal lesions undergoing a 12-week structured rehabilitation program including swimming. Group-level analyses suggested that horses with cervical lesions may exhibit slightly greater flexion–extension range of motion than those with thoracolumbar lesions, but this trend was based on a small and unbalanced sample and should be interpreted with caution. The main finding remains that swimming, within a structured rehabilitation program, did not induce substantial or consistent changes in thoracolumbar mobility. Within-group analyses showed that only the thoracolumbar group exhibited a significant phase effect, characterized by a slight decrease in thoracolumbar mobility during the second month of aquatic training (AT2), whereas no significant change was observed after the first month (AT1). These variations remained small, with effect sizes below conventional thresholds for practical relevance.

At the individual level, considerable variability was observed, with both increases and decreases in thoracolumbar motion, regardless of lesion location. These heterogeneous responses underscore that spinal mobility does not evolve uniformly under rehabilitation, whether or not swimming is included, and highlight the need for individualized follow-up.

From a clinical perspective, swimming appears feasible within structured rehabilitation programs but does not induce substantial, consistent changes in thoracolumbar kinematics at the population level. Objective, sensor-based motion analysis provides a valuable complement to clinical assessment, enabling the detection of subtle functional adaptations that may not be apparent through qualitative evaluation alone.

Further work, already underway, will integrate concurrent clinical and imaging data (including radiography, ultrasonography, and scintigraphy) to clarify the relationships between lesion dynamics, pain expression, and kinematic outcomes. Together, these complementary analyses will contribute to refining rehabilitation strategies and improving the individualized management of horses with axial musculoskeletal disorders.

## Figures and Tables

**Figure 1 animals-16-00103-f001:**
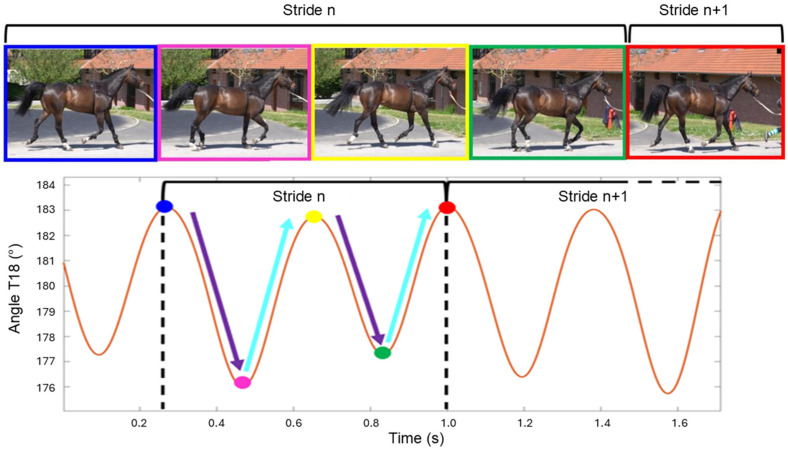
Illustration of flexion–extension movements (T18 angle) at the thoracolumbar junction during two consecutive strides. The orange line represents the IMU-derived angle over time. Blue and yellow points mark flexion maxima during swing phases; pink and green points mark extension minima during stance phases. The red point corresponds to the first flexion peak of the subsequent stride. Purple and cyan arrows represent, respectively, the two extension and two flexion movements occurring within a single stride.

**Table 1 animals-16-00103-t001:** Distribution of horses included in the study and associated lesions. SF: French warmblood; PFS: French pony; Z: Zangersheide; SJ: show jumping; OA: osteoarthritis; AP-SIVJ: articular processes-synovial intervertebral joints; DSP: dorsal spinal process; C: cervical vertebra; T: thoracic vertebra; R: rib; L: lumbar vertebra.

Horse	Sex	Breed	Discipline	Age (y)	Height (cm)	Body Mass (kg)	Main Lesions (Location)
**Cervical Group (n = 4)**							
#02	Mare	SF	SJ	14	160	502	OA AP-SIVJ (C5–T1)
#06	Gelding	SF	Eventing	10	166	566	Vertebral symphysis abnormality (C6–C7), OA AP-SIVJ (C6–T1), R1 hypogenesia
#14	Gelding	SF	SJ	9	177	596	OA AP-SIVJ (C6–C7), vertebral symphysis abnormality (C6–T1)
#19	Gelding	SF	SJ	5	163	528	OA AP-SIVJ (C6–C7), vertebral symphysis abnormality (C6–C7)
**Thoracolumbar Group (n = 12)**							
#01	Mare	SF	SJ	11	164	504	OA AP-SIVJ (T15–T16), DSP impingement (T13–T14, T16–T17)
#03	Gelding	PFS	SJ	12	148	424	OA AP-SIVJ (T15–T16, T18–L1 > L1–L4), DSP impingement (T15–T18)
#09	Gelding	SF	SJ	11	170	618	Vertebral body dysplasia (T9–T10 cuneiformis vertebrae), curvature abnormality (thoracic lordosis), DSP impingement (T12–T16)
#11	Gelding	SF	SJ	13	166	594	OA AP-SIVJ (T14–T18), DSP impingement (T13–T17)
#12	Gelding	SF	SJ	11	170	520	DSP impingement (T13–T18), OA AP-SIVJ (T14–T17)
#13	Gelding	SF	SJ	10	177	562	OA AP-SIVJ (L1–L3)
#15	Mare	SF	SJ	8	166	560	OA AP-SIVJ (T18–L2), curvature abnormality (lumbar kyphosis + scoliosis)
#17	Gelding	SF	SJ	9	180	616	Vertebral symphysis abnormality (L6–S1)
#18	Gelding	SF	SJ	5	167	580	Curvature abnormality (thoracolumbar lordo-kiphosis + lumbar scoliosis), OA AP-SIVJ (T15–T18)
#20	Gelding	SF	SJ	5	167	550	OA AP-SIVJ (T15–L2), DSP impingement (T18–L1)
#21	Mare	SF	SJ	8	166	524	Ventral spondylosis (T12–T13), DSP impingement (T14–L2), OA AP-SIVJ (T15–T16)
#23	Mare	Z	SJ	9	168	564	Ventral spondylosis (T11–T12)

**Table 2 animals-16-00103-t002:** Weekly training schedules during Phase 1 (land training) and Phase 2 (aquatic training). “jumping”: ridden session focused on jumping exercises, including the sequence of a show jumping course with 20–25 jumps and a max high of 1 m10; “gallop”: session with intervals of canter between 28–30 km/h on the racetrack; “gymnastic-cavaletti”: mixed session combining flatwork and gymnastic exercises over cavaletti (<50 cm); “Rest day”: Sunday dedicated to light exercise only, consisting of walking in a motorized horse walker (no ridden work); each riding session, except for the gallop session, lasted 40 to 50 min and was preceded by a period on a motorized horse walker.

Day	Phase 1—Land Training (4 Weeks)	Phase 2—Aquatic Training (8 Weeks)
Monday	Walker (40 min) + flatwork (15 min walk, 4 min slow canter, 30 min flatwork at 3 gaits)	Walker (40 min) + swimming (see Table 3)
Tuesday	Walker (40 min) + gymnastic-cavaletti (15 min walk, 4 min slow canter, 30 min work at 3 gaits with cavaletti <50 cm)	Walker (40 min) + gymnastic-cavaletti (same as Phase 1)
Wednesday	Walker (40 min) + gallop (15 min walk, 4 min slow canter, 2 min walk, 3 × 3 min gallop + 2 min trot)	Walker (40 min) + swimming (see Table 3)
Thursday	Walker (40 min) + jumping (15 min walk, 4 min slow canter, 30 min work at 3 gaits with jumping: 20–25 jumps approx. 1 m)	Walker (40 min) + jumping (same as Phase 1)
Friday	Walker (40 min) + flatwork (5 min walk, 4 min slow canter, 30 min flatwork at 3 gaits)	Walker (40 min) + swimming (see Table 3)
Saturday	Walker (30 min) + lunge work (2 min walk, 4 min slow canter, 2 min slow trot, 2 min medium trot, 3 min medium canter, 1 min walk, 3 min medium canter, 3 min medium trot, 5 min walk)	Walker (30 min) + lunge work (same as Phase 1)
Sunday	Rest day + walker (30 min)	Rest day + walker (30 min)

**Table 3 animals-16-00103-t003:** Swimming workload during Aquatic Training 1 (AT1; weeks 5–8) and Aquatic Training 2 (AT2; weeks 9–12). Distances swum per session (m) and total session duration (min), including swimming, inter-lap breaks, and active recovery phases, are reported for each training day (Monday, Wednesday, Friday). Weekly total swimming distances are provided for each week. No swimming session was performed on Friday of week 12 (-) due to the final scintigraphic examination.

Phase	Week	Monday		Wednesday		Friday		Total Weekly Distance (m)
		Distance Swum (m)	Session Duration (min)	Distance Swum (m)	Session Duration (min)	Distance Swum (m)	Session Duration (min)	
Aquatic training 1	W5	240	10	330	12	330	12	900
	W6	440	22	440	22	440	22	1320
	W7	605	28	550	23	715	31	1870
	W8	605	28	660	27	770	32	2035
Aquatic training 2	W9	660	27	605	28	770	32	2035
	W10	770	32	660	27	935	36	2365
	W11	770	32	660	27	935	36	2365
	W12	770	32	605	28	-	-	1375

**Table 4 animals-16-00103-t004:** Individual values and group means (±SD) of T18 flexion–extension range of motion angle (ROM, °) and stride frequency (Std-Freq, std/s) across the three training phases for horses presenting either cervical or thoracolumbar lesions. The three phases are as follows: land training (LT), aquatic training 1 (AT1), and aquatic training 2 (AT2).

Horse ID	LT		AT1		AT2	
	ROM (°)	Std-Freq (std/s)	ROM (°)	Std-Freq (std/s)	ROM (°)	Std-Freq (std/s)
**Cervical group (n = 4)**						
#02	4.5 ± 0.7	1.42 ± 0.06	4.1 ± 0.5	1.42 ± 0.05	4.2 ± 0.7	1.39 ± 0.04
#06	3.7 ± 0.4	1.42 ± 0.05	4.0 ± 0.4	1.45 ± 0.05	3.8 ± 0.6	1.43 ± 0.05
#14	5.0 ± 0.9	1.27 ± 0.09	5.2 ± 0.5	1.27 ± 0.03	5.3 ± 0.7	1.30 ± 0.12
#19	4.5 ± 0.7	1.36 ± 0.25	4.2 ± 1.1	1.41 ± 0.06	4.0 ± 0.6	1.38 ± 0.06
**Group mean**	**4.4 ± 0.8**	**1.37 ± 0.14**	**4.4 ± 0.9**	**1.39 ± 0.08**	**4.3 ± 0.8**	**1.38 ± 0.08**
**Thoracolumbar group (n = 12)**						
#01	3.5 ± 0.4	1.37 ± 0.06	2.6 ± 0.5	1.41 ± 0.06	3.0 ± 0.3	1.39 ± 0.05
#03	3.0 ± 1.0	1.47 ± 0.07	3.2 ± 0.5	1.46 ± 0.06	3.1 ± 0.9	1.49 ± 0.11
#09	3.2 ± 0.5	1.37 ± 0.05	4.0 ± 0.4	1.35 ± 0.05	3.6 ± 0.5	1.35 ± 0.05
#11	3.6 ± 0.5	1.29 ± 0.05	3.6 ± 0.6	1.29 ± 0.05	3.7 ± 0.6	1.29 ± 0.05
#12	3.9 ± 0.6	1.43 ± 0.06	4.1 ± 0.5	1.40 ± 0.04	3.9 ± 0.5	1.41 ± 0.05
#13	3.4 ± 0.5	1.30 ± 0.04	3.3 ± 0.4	1.31 ± 0.06	3.4 ± 0.8	1.30 ± 0.05
#15	3.5 ± 0.5	1.36 ± 0.06	3.7 ± 0.4	1.33 ± 0.04	3.8 ± 0.5	1.32 ± 0.06
#17	3.9 ± 0.6	1.32 ± 0.05	4.1 ± 0.6	1.34 ± 0.05	3.2 ± 0.7	1.38 ± 0.05
#18	2.5 ± 0.6	1.28 ± 0.10	2.2 ± 0.5	1.32 ± 0.07	2.3 ± 0.7	1.37 ± 0.13
#20	4.4 ± 1.0	1.36 ± 0.08	4.4 ± 0.8	1.37 ± 0.11	3.4 ± 0.6	1.38 ± 0.14
#21	2.2 ± 0.6	1.40 ± 0.08	2.4 ± 0.5	1.40 ± 0.06	2.5 ± 0.9	1.38 ± 0.06
#23	3.5 ± 0.5	1.45 ± 0.06	3.5 ± 0.6	1.45 ± 0.06	3.4 ± 0.6	1.49 ± 0.09
**Group mean**	**3.4 ± 0.9**	**1.37 ± 0.09**	**3.4 ± 0.8**	**1.37 ± 0.08**	**3.3 ± 0.8**	**1.39 ± 0.10**

**Table 5 animals-16-00103-t005:** Results of linear mixed-effects models (type III ANOVA) and post hoc Tukey–Kramer (TK) tests assessing changes in T18 flexion–extension range of motion (°) across the three training phases for each horse and globally for each group. Mean differences (Δ) between phases are expressed in degrees (°), together with percentage of change (% evol), associated TK *p*-values, and Cohen’s d effect sizes (C’s d). Comparisons include the following: AT1-LT (change from land training to aquatic training 1); AT2-AT1 (change from aquatic training 1 to aquatic training 2); and AT2-LT (change from land training to aquatic training 2). Negative values indicate a decrease in thoracolumbar flexion–extension range of motion over time, whereas positive values indicate an increase. Effect sizes are omitted (‘-’) when non-significant or when variance was insufficient for calculation.

Horse		Δ AT1-LT (°)			Δ AT2-AT1 (°)			Δ AT2-LT (°)		
	Type III ANOVA	Estimate (°) (% evol)	TK	C’s d	Estimate (°) (% evol)	TK	C’s d	Estimate (°) (% evol)	TK	C’s d
**Cervical group (n = 4)**										
#02	<0.001	−0.3 (−7%)	0.001	0.5	0 (0%)	0.963	-	−0.3 (−7%)	<0.001	0.3
#06	<0.001	0.4 (11%)	<0.001	0.8	−0.2 (−5%)	<.001	0.4	0.1 (3%)	0.027	0.3
#14	<0.001	0.3 (6%)	0.001	0.3	0.2 (4%)	0.0320	0.1	0.5 (10%)	<0.001	0.4
#19	<0.001	−0.3 (−7%)	0.126		−0.4 (−10%)	<0.001	0.2	−0.7 (−16%)	<0.001	0.8
**Mean**	**0.092**	**0.1 (2%)**	**-**	**-**	**−0.1 (−2%)**	**-**	**-**	**0 (0%)**	**-**	**-**
**Thoracolumbar group (n = 12)**										
#01	<0.001	−0.7 (−20%)	<0.001	1.9	0.3 (12%)	<0.001	0.8	−0.5 (−14%)	<0.001	1.4
#03	0.102	0.1 (3%)	-	-	0 (0%)	-	-	0.2 (6%)	-	-
#09	<0.001	0.8 (25%)	<0.001	1.8	−0.4 (−10%)	<0.001	1	0.3 (9%)	<0.001	0.8
#11	0.338	−0.1 (−3%)	-	-	0.1 (3%)	-	-	0 (0%)	-	-
#12	0.172	0.1 (3%)	-	-	−0.1 (−2%)	-	-	0 (0%)	-	-
#13	0.366	−0.1 (−3%)	-	-	0.1 (3%)	-	-	0 (0%)	-	-
#15	0.023	0.1 (3%)	0.418	-	0.1 (3%)	0.267	-	0.1 (3%)	0.017	0.6
#17	<0.001	0.3 (12%)	0.003	0.3	−0.8 (−20%)	<0.001	1.4	−0.6 (−15%)	<0.001	1.1
#18	<0.001	−0.3 (−12%)	<0.001	0.5	0.1 (5%)	0.456	-	−0.2 (−8%)	0.036	0.4
#20	<0.001	0 (0%)	0.975	-	−0.8 (−18%)	<0.001	1.2	−0.8 (−18%)	<0.001	1.1
#21	0.004	0.2 (10%)	0.170	-	0.1 (4%)	0.301	-	0.3 (14%)	0.003	0.4
#23	0.500	−0.1 (−3%)	-	-	0 (0%)	-	-	0 (0%)	-	-
**Mean**	**<0.001**	**0 (0%)**	**0.199**	**-**	**−0.1 (−3%)**	**<0.001**	**0.2**	**−0.1 (−3%)**	**0.002**	**0.1**

## Data Availability

Additional data are available upon reasonable request from the corresponding author.

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
