# Peer review of "Group and Individual Changes in Spinal Mobility During a 12-Week Rehabilitation Program Including Swimming in Horses with Axial Musculoskeletal Lesions"

_animals, 2025, doi:10.3390/ani16010103_

Round 1
Reviewer 1 Report
Comments and Suggestions for Authors
Review of Manuscript
Group and individual changes in spinal mobility during a 12-2 week rehabilitation program including swimming in horses 3 with axial musculoskeletal lesions
General
A nice and interesting study on back movement. I have made detailed comments below.
Overall I have a couple of comments, the first one being related to the emphasis of ‘individual horses’ over group-level effects. My recommendation would be to reduce the emphasis of effects reported in individual horses and concentrate on the group level effects.
Second, the absence of a control group should be mentioned in the abstract. The study design does not really allow to draw conclusions about the presence/absence of swimming exercise on the reported changes over time without a control group. This is of course a common issue with studies implemented in clinical practice and the authors mention this in the discussion. It would appear fair to mention this more prominently for example in the abstract and in my opinion the absence of a control group does not diminish the value of the present study.
For me it is very surprising that the authors, who have published a number of manuscripts on movement symmetry in horses, have restricted the analysis of back movement to straight line trot. The authors alos state that horses with limb related issues were excluded from the study, hence it would be even more interesting to provide evidence about the locomotor asymmetry status of the horses included in the study.
Given that the study has concentrated on straight-line trot and utilized sensors mounted over the withers and over the tubera sacrale, which are regularly used for movement symmetry analysis, I would have expected some quantitative evidence of ‘movement symmetry’ for the horses enrolled in the study. These data should be made available (at the very least as supplementary material) so that future studies can utilize this information for example as part of a meta-analysis. Multiple previous studies have provided thoraco-lumbo-sacral ranges of motion (in three or six dimensions) and typically movement symmetry variables are reported at least as a means of objective, quantitative overview or have directly associated thoraco-lumbo-sacral ranges of motion with movement symmetry.
A more minor point: what is the reason for restricting back movement analysis to trot where the flexion-extension range is typically rather restricted due to the 'dynamic stability' required to move in a manner where diagonal pairs of limbs are contacing the ground simultaneously?
Congratulations to a very nice study!
Detailed comments
Simple summary:
generally appropriately written with more accessible language. In particular for the ‘simple summary’, I would love to see a little less ‘emphasis’ of the individualized results, i.e. some horses showing increased, some decreased back mobility since in particular for a more ‘lay audience’ of the simple summary, this may detract too much from the more robust statistical results. Please consider deemphasizing the individual results.
Abstract:
Similar comments here but appreciate that a scientific audience will hopefully have a better grasp of the concept. Generally I would recommend to shift the focus of the abstract a little more onto the statistically robust results of the present study (and away from the individual horses and also away from the ‘additional studies’.
Please add some ‘numbers’ to the abstract. Representative values for the changes (or lack thereof) measured in the present study.
Introduction:
Curious choice of references utilized for validation studies of IMUs in the context of ‘back movement’ (“Several studies have demonstrated agreement between MOCAP and 76 IMU measurements, without proportional bias [24, 26].”). In particular refence 26 seems like a non-obvious choice judging from the title of the manuscript. I failed to be able to locate a manuscript by that title and additionally the DOI link seems to link to a different EVJ manuscript. Please double-check this reference and there are possibly other more relevant publications on a comparison between IMUs and motion capture in the context of thoraco-lumbar movement. Reference 24 certainly makes a lot of sense.
Line 82: the word ‘evolution’ might possibly not be the most suitable choice? Maybe something like ‘longitudinal development’ or ‘changes over time’ might be more specific in the context of describing changes in back mobility over a time period.
Line 80-94: a couple of observations here.
Maybe add the term ‘flexion-extension’ somewhere since from reading the Hatrisse validation study, that validation was focused on flexion-extension specifically to make a distinction between other methods (mocap or IMU based) that assess multiple ‘planes’ (translational and/or rotational).
I am intrigued by the hypothesis: how do you go about testing that something ‘changes minimally’?
Can you provide more logic for ‘not testing’ for differences between horses with lesions in different anatomical regions? One of the questions indicates that this is of interest, the hypothesis does not. Do you need to add a second hypothesis?
Materials and methods
Line 97-105. I would suggest adding a ‘pointer’ to Table 1 in this section, which nicely lists the different horses. Would be lovely to have information about the height (and/or body mass) too.
Line 106-115: since horses with limb related pathologies were excluded it would be super nice to learn more about whether for the inclusion/exclusion decision, any quantitative gait information had been used?
Line 131-138: I guess now it becomes a little clearer why you were maybe not expecting ‘more pronounced’ changes in association with the exercise protocol given that in particular the thoracolumbar group could include a range of different abnormalities. Still it is not clear to me how you test for ‘minimal changes’?
Line 144: what is a ‘mechanical walker’. As opposed to what different type of walker?
Line 146-157: Please refer the reader to Table 2. Is it possible to indicate an average (range) for duration of training sessions?
Line 156-157: please add more detail: were the horses instrumented during this phase and spinal mobility measured? Ridden? Gait? Etc
Line 178-183: please provide additional details about the exercise: 50m in one go? Or were multiple ‘trotups’ used when necessary. Horses were assessed in-hand presumably. Was the same handler used or similar ‘equipment’ (head collar, bridle … ).
Line 184-193: after re-reading the validation manuscript, I have a question here about the underlying method that calculates angles. If I understand correctly (and please do correct me anywhere I am wrong), the angles are calculated from ‘vertical displacement traces’ of the three sensors. How does the length of the back go into these calculations? The angles between the ‘vertical displacements’ would change according to the horizontal distance between the sensors. Likely not majorly so in the range of expected variation of ‘back length’ in the horse. They would also change (again only slightly) as a function of the vertical position of the three sensors relative to each other, for example in a horse with a more extended back or a horse with a more flexed back the same vertical displacements would result in (slightly) different angle calculations. Were additional measurements such as the horizontal and vertical distance between the sensors in the standing (or moving) horse taken into account for the angle calculations?
Line 191-192: it would be super nice to illustrate these variables in an example trace. I understand that this refers to a published manuscript, nonetheless I think the reader would benefit from understanding the variables used. In particular, what are the ‘mean flexion and extension amplitudes’. Does this refer to each amplitude separately or is the ‘amplitude from maximal flexion to maximal extension? What is ‘neutral’ (and how is this defined with/without additional knowledge about the relative positioning of the three sensors)?
Line 207-212: did all horses have the same number of strides included in the analysis? I am not a statistics expert, so have a coupe questions here: are multiple strides really ‘random’? Movement dynamics over series of strides would likely mean that what happens during one stride is highly dependent on what has happened in the previous stride? I am also wondering how stride frequency was entered into this model? Was it used as a fixed covariate’ and then the model evaluated at a ‘mean stride frequency’? Could you provide stride frequency values for each horse and phase (Table 4)?
Results
Line 224-237: I wonder whether it would make some sense to provide some example traces of T18 flexion-extension angle traces.
Table 5: would be super to explicitly state whether positive or negative changes refer to increases or decreases inn back movement over time.
Line 272: replace the word ‘evolution’ as suggested initially
Line 272-316: as mentioned in one of the initial comments, it is my opinion that while variation between horses is possibly interesting, this aspect should not be over-emphasized. Overall, average effects should be emphasized.
Discussion
Line 331-333: or in addition to head/neck position the “direct influence” of “back posture” for example through use of training aids including elastic resistance band usage.
Line 334-343: what about dynamic stability as an explanation? I always find this puzzling: is ‘reduced’ back movement a good or a bad thing in particular in trot with diagonal pairs of limbs in contact with the ground simultaneously.
Line 359-376: maybe an opportunity to comment again on stride frequency (or maybe this is better done in the subsequent section on ‘methodological’ implications.
Line 379-384: in particular the absence of a control group should maybe be mentioned in the abstract.
Line 387-389: not sure I understand what is meant by ‘global’ here? Please clarify.
Line 389-390: possibly more detail needed to clarify the point about ‘sensor misalignment’? Only people with intricate knowledge of the underlying technique (with reference to the questions above about additional measurements about (relative) position of the three sensors?) will be able to appreciate the comment here without further explanations.
Line 390-391: when you refer to angular resolution, is this with reference to the individual sensors or the three-sensor method used here?
Line 395-397: it would be great to make the fact that stride frequency was used as a covariate more prominent throughout the manuscript. This is a strong point of the analysis. At what mean stride frequency were the models analyzed?
Line 402-405: I feel that this section could do with some references to published work on EMG of back muscles and to references to manuscripts that provide quantitative evidence about back movement across several landmarks, independent of what kinematic techniques have been used.
Reviewer 2 Report
Comments and Suggestions for Authors
Material and methods as well as the results were well described and discussed, the latter including the limitations of the study. Some of these make it senseful to add to the title “Preliminary results on…..”
I have few queries, suggestions and corrections:
- Describe the swimming pool (oval, circular, straight; length; water depth; guided or not, etc.)
- Which was the basis to define the swimming distances?
- Please add range of swimming duration for the different distances
- For some reason line numbers 262-270 are visible on the left side of the page, mostly inside the table. Please correct.
- From the discussion it appears that a positive effect of the reha measures would have been an increased range of motion of the thoraco-lumbar junction. The only significant result found, albeit small only, was a decrease of it. Does this indicate that swimming might deteriorate the mobility of the region described in this manuscript?
- What did owners / riders of horses report after the reha-period?
Round 2
Reviewer 1 Report
Comments and Suggestions for Authors
Thank you very much for a very thorough revision of your manuscript. Your responses and revisions are much appreciated.
line 178: 'excepted' should be 'except'
Appendix A: Horse #3 has a suspiciously high pelvic movement asymmetry value of 77.3% I suspect this is a typo.
Author Response
Thank you very much for a very thorough revision of your manuscript. Your responses and revisions are much appreciated.
We would like to sincerely thank the reviewer for the careful reading of the revised manuscript and for their positive and encouraging feedback. The manuscript has been substantially improved thanks to the reviewer’s detailed, thoughtful, and constructive comments, and we are truly grateful for the time and attention devoted to this review.
line 178: 'excepted' should be 'except'
We thank the reviewer for pointing this out. The word “excepted” has been corrected to “except” in the revised manuscript.
Appendix A: Horse #3 has a suspiciously high pelvic movement asymmetry value of 77.3% I suspect this is a typo.
We are very grateful to the reviewer for this vigilant observation. This was indeed a typographical error resulting from an incorrect placement of the decimal point. The correct value is 7.7%, not 77.3%. We have corrected this error in Appendix A. Fortunately, as the reviewer rightly notes, a horse presenting such a high asymmetry value would not have met the inclusion criteria for this study.
Once again, we thank the reviewer for this attentive and careful review, which has greatly contributed to improving the accuracy and overall quality of the manuscript.